

# Strain-level genetic diversity of *Methylophaga nitratireducenticrescens* confers plasticity to denitrification capacity in a methylotrophic marine denitrifying biofilm

Valérie Geoffroy[1,2], Geneviève Payette[1], Florian Mauffrey[3], Livie Lestin[1], Philippe Constant[1] and Richard Villemur[1]

[1] Institut Armand-Frappier, Institut National de la Recherche Scientifique, Laval, Québec, Canada
[2] Lallemand, Montreal, Québec, Canada
[3] Laboratoire de santé publique du Québec, Ste-Anne-de-Bellevue, Québec, Canada

Corresponding author
Richard Villemur,
richard.villemur@iaf.inrs.ca

## ABSTRACT

**Background**. The biofilm of a methanol-fed, fluidized denitrification system treating a marine effluent is composed of multi-species microorganisms, among which *Hyphomicrobium nitrativorans* NL23 and *Methylophaga nitratireducenticrescens* JAM1 are the principal bacteria involved in the denitrifying activities. Strain NL23 can carry complete nitrate ($NO_3^-$) reduction to $N_2$, whereas strain JAM1 can perform 3 out of the 4 reduction steps. A small proportion of other denitrifiers exists in the biofilm, suggesting the potential plasticity of the biofilm in adapting to environmental changes. Here, we report the acclimation of the denitrifying biofilm from continuous operating mode to batch operating mode, and the isolation and characterization from the acclimated biofilm of a new denitrifying bacterial strain, named GP59.

**Methods**. The denitrifying biofilm was batch-cultured under anoxic conditions. The acclimated biofilm was plated on *Methylophaga* specific medium to isolate denitrifying *Methylophaga* isolates. Planktonic cultures of strains GP59 and JAM1 were performed, and the growth and the dynamics of $NO_3^-$, nitrite ($NO_2^-$) and $N_2O$ were determined. The genomes of strains GP59 and JAM1 were sequenced and compared. The transcriptomes of strains GP59 and JAM1 were derived from anoxic cultures.

**Results**. During batch cultures of the biofilm, we observed the disappearance of *H. nitrativorans* NL23 without affecting the denitrification performance. From the acclimated biofilm, we isolated strain GP59 that can perform, like *H. nitrativorans* NL23, the complete denitrification pathway. The GP59 cell concentration in the acclimated biofilm was 2–3 orders of magnitude higher than *M. nitratireducenticrescens* JAM1 and *H. nitrativorans* NL23. Genome analyses revealed that strain GP59 belongs to the species *M. nitratireducenticrescens*. The GP59 genome shares more than 85% of its coding sequences with those of strain JAM1. Based on transcriptomic analyses of anoxic cultures, most of these common genes in strain GP59 were expressed at similar level than their counterparts in strain JAM1. In contrast to strain JAM1, strain GP59 cannot reduce $NO_3^-$ under oxic culture conditions, and has a 24-h lag time before growth and $NO_3^-$ reduction start to occur in anoxic cultures, suggesting that both strains regulate differently the expression of their denitrification genes. Strain GP59 has the ability to

reduce $NO_2^-$ as it carries a gene encoding a NirK-type $NO_2^-$ reductase. Based on the CRISPR sequences, strain GP59 did not emerge from strain JAM1 during the biofilm batch cultures but rather was present in the original biofilm and was enriched during this process.

**Discussion**. These results reinforce the unique trait of the species *M. nitratireducenticrescens* among the *Methylophaga* genus as facultative anaerobic bacterium. These findings also showed the plasticity of denitrifying population of the biofilm in adapting to anoxic marine environments of the bioreactor.

# INTRODUCTION

Denitrification describes the successive reduction of nitrate ($NO_3^-$) to nitrite ($NO_2^-$), nitric oxide (NO), nitrous oxide ($N_2O$), and nitrogen ($N_2$) (*Van Spanning, Delgado & Richardson, 2005*). This process is used by bacteria for respiration in environments with low oxygen concentrations and with $NO_3^-$, $NO_2^-$, NO, and $N_2O$ as electron acceptors. The process is driven by metalloenzymes $NO_3^-$ reductase, $NO_2^-$ reductase, NO reductase, and $N_2O$ reductase (*Einsle & Kroneck, 2004*). As a facultative trait, denitrification occurs frequently across environments, and is performed by bacteria of diverse origins (*Zumft, 1997*). Furthermore, numerous microorganisms carry incomplete denitrification pathways for either their growth or detoxification, as $NO_2^-$ and NO are deleterious molecules (*Kaspar, 1982*; *Poole, 2005*; *Schreiber et al., 2012*; *Simon & Klotz, 2013*).

Several studies were carried out in our laboratory on a naturally-occurring multispecies denitrifying biofilm that has developed in a methanol-fed, fluidized denitrification continuous system that treated recirculating water of a marine aquarium at the Montreal Biodome. The biofilm consists of at least 15 bacterial species and numerous protozoans (*Labbé et al., 2003*; *Laurin et al., 2008*), among which *Methylophaga* spp. and *Hyphomicrobium* spp. compose more than 50% of the biofilm (*Labbé et al., 2007*). *Rissanen et al. (2016)* also observed the combination of *Methylophaga* spp. and *Hyphomicrobium* spp. in the fluidized-bed type denitrification reactors treating the recirculating seawater of the public fish aquarium SEA LIFE at Helsinki, Finland, suggesting the importance of these two genera in marine denitrification processes.

We isolated *Hyphomicrobium nitrativorans* NL23 and *Methylophaga nitratireducenticrescens* JAM1 from the biofilm of the Biodome denitrification system and showed they are the main actors of the denitrifying activities (*Labbé et al., 2003*; *Auclair et al., 2010*; *Auclair, Parent & Villemur, 2012*; *Martineau et al., 2013a*; *Villeneuve et al., 2013*). *H. nitrativorans* NL23 can carry out complete $NO_3^-$ reduction to $N_2$, whereas *M. nitratireducenticrescens* JAM1 can perform three out of the four reduction steps; it misses the reduction of $NO_2^-$ to NO (*Martineau, Mauffrey & Villemur, 2015*; *Martineau et al., 2013b*; *Mauffrey et al., 2017*; *Mauffrey, Martineau & Villemur, 2015*; *Villeneuve et al., 2012*). In addition to
*H. nitrativorans* NL23 and *M. nitratireducenticrescens* JAM1, we have demonstrated that a small proportion of other denitrifiers exists in the biofilm of the Biodome denitrification system (*Auclair, Parent & Villemur, 2012*), suggesting the potential plasticity of the biofilm in adapting to environmental changes.

*Methylophaga* species are halophilic marine methylotrophic gammaproteobacteria that do not grow in the absence of NaCl and use one-carbon compounds (e.g., methanol) as sole carbon and energy sources with carbon assimilation proceeding via the 2-keto-3-deoxy-6-phosphogluconate (KDPG)-variant ribulose monophosphate (RuMP) pathway (*Villeneuve et al., 2013*; *Boden, 2012*). The operating conditions of the Biodome denitrification process led to the enrichment of *M. nitratireducenticrescens* JAM1 in the biofilm, which is the only reported isolated *Methylophaga* species that can grow under anoxic conditions with $NO_3^-$ or $N_2O$ as sole electron acceptors (*Auclair et al., 2010*; *Mauffrey et al., 2017*; *Mauffrey, Martineau & Villemur, 2015*). This trait is correlated with the presence in the genome of two gene clusters encoding dissimilatory $NO_3^-$ reductases (*narGHJI*, referred as *nar1* and *nar2*), two gene clusters encoding cytochrome *bc*-type complex NO reductases (referred as *cnor1* and *cnor2*) and one gene cluster encoding a dissimilatory $N_2O$ reductase (*nos*) (*Villeneuve et al., 2013*; *Mauffrey, Martineau & Villemur, 2015*). *M. nitratireducenticrescens* JAM1 lacks a gene encoding a dissimilatory copper- (NirK) or cytochrome cd1-type (NirS) $NO_2^-$ reductase.

Our group has also investigated the potential of using the batch operating mode for the denitrification system instead of the continuous one as it prevailed at the Montreal Biodome. With this mode, a better real-time control was obtained to avoid sulfate-reduction that can be easily occurring in seawater biotreatments (*Labelle, 2004*). In this report, we aim to assess the impact of the batch operating mode on the bacterial population of the denitrifying biofilm. We observed the disappearance of *H. nitrativorans* NL23 without the loss of the denitrifying activities during the biofilm acclimation to this mode. We hypothesized that new denitrifiers were enriched during the biofilm acclimation, which displaced *H. nitrativorans* NL23. From this biofilm, a new denitrifying bacterial strain, named GP59, was isolated that is closely related to *M. nitratireducenticrescens* JAM1. We compared strain GP59 with *M. nitratireducenticrescens* JAM1 at the physiological, genomic and transcriptomic levels. Genomic data were also compared with available genomes of *Methylophaga* species. Our results present new insights of *M. nitratireducenticrescens* under anoxic environments. They also provide indications that strain-level genomic heterogeneities can influence the dynamism of the biofilm microbial population in adapting to new operating conditions.

## MATERIAL AND METHODS

### $NO_3^-$, $NO_2^-$ and $N_2O$ measurements

Measurements of $NO_3^-$, $NO_2^-$ and $N_2O$ concentrations in all biofilm and planktonic cultures were performed using ion chromatography as described in *Mauffrey et al. (2017)*.

## Biofilm acclimation

Artificial seawater (ASW) medium was composed of (for 1 liter solution): 27.5 g NaCl, 10.68 g $MgCl_2$*$6H_2O$, 2 g $MgSO_4$*$7H_2O$, 1 g KCl, 0.5 g $CaCl_2$, 456 μL of $FeSO_4$*$7H_2O$ 4 g/L, 5 mL of $KH_2PO_4$ 51.2 g/L, 5 mL of $Na_2HPO_4$ 34 g/L. It was supplemented with 1 mL of trace elements (master solution: $FeSO_4$*$7H_2O$ 0.9 g/L, $CuSO_4$*$5H_2O$ 0.03 g/L and $MnSO_4$*$H_2O$ 0.2 g/L) (*Labbé, Parent & Villemur, 2003*) and $NaNO_3$ (21.4 mM or 300 mg-N/L). The pH was adjusted (NaOH) at 8.0. Three 120-mL vials containing each 20 unused "Bioflow 9 mm" carriers (Rauschert, Steinwiessen, Germany) and 60 mL of ASW were purged of oxygen for 10 min with pure nitrogen gas ($N_2$, purity > 99.9%; Praxair, Mississauga, ON, Canada), sealed with sterile septum caps and autoclaved. Prior use, these carriers were washed with HCl 10% (v/v) for 3 h, rinsed with water and autoclaved. A 90 μL volume (72 mg) of filtered-sterilized methanol was then added to the vials (0.15% v/v final concentration).

Carriers (Bioflow 9 mm) with the denitrifying biofilm were taken from the denitrification system at the Montreal Biodome and frozen at −20 °C in seawater with 20% glycerol (*Laurin et al., 2006*) until use. The biomass from several frozen carriers was thawed, scrapped, weighted and dispersed in the ASW medium at 0.08 g (wet weight)/mL. The biomass (5 mL/vial, 0.4 g of biofilm) was then distributed with a syringe and an $18_G1\frac{1}{2}$ needle to three vials. The vials were incubated at room temperature and 100 rpm with an orbital shaker in the dark. In average once a week, the vial was opened in ambient air, the carriers were taken, gently washed with ASW medium to remove the excess medium and the free bacteria, then transferred into vials with fresh anoxic medium, and incubated in the same conditions (Fig. S1). Methanol and $NO_3^-$ were added when needed if $NO_3^-$ was completely depleted during the week. After the 5th transfer, the biofilm was collected. One part was preserved in 15% glycerol at −70 °C for further used such as DNA extraction (see below). The other part was used to isolate bacterial isolates (next subsection).

## Isolation of Methylophaga sp. strain GP59

The acclimated biofilm was dispersed in saline solution (3% NaCl, 34.2 mM phosphate buffer pH 7.4), and serial dilutions were made and inoculated onto the *Methylophaga* medium 1403 (American Type Culture Collection [ATCC], Manassas, VA, USA) supplemented with 1.5% agar and 0.3% v/v methanol. The plates were incubated at 30 °C for no more than 7 days, under oxic conditions (no $NO_3^-$) or under anoxic conditions in anoxic jars with a gaspak (BBL™ GasPak™ Plus Anaerobic System). For the latter conditions, the medium was supplemented with 21.4 mM $NaNO_3$. Several colonies were picked and cultured in the *Methylophaga* 1403 liquid medium under oxic or anoxic conditions for 2–7 days. After restreaking three times onto agar medium, isolates were tested for growth under anoxic conditions. $NO_3^-$ and $NO_2^-$ consumption was measured after four days of incubation. Identification of the isolates was determined by extracting total DNA from the isolates, followed by PCR amplification of the 16S ribosomal RNA (rRNA) genes (forward primer 5′-AGAGTTTGATCCTGGCTCAG-3′ and reverse primer 5′-AAGGAGGTGATCCAGCCGCA-3′, which correspond to positions 8 to 27 and 1,521

to 1,540 in the *Escherichia coli* 16S rRNA gene) and the sequencing of the resulting PCR products (McGill University and Genome Quebec Innovation Center, Montréal QC, Canada).

## Planktonic pure cultures

*M. nitratireducenticrescens* GP59 and JAM1 were cultured in the *Methylophaga* medium 1403 as described by *Villeneuve et al. (2013)* with 0.3% methanol. When required, $NO_3^-$ ($NaNO_3$) was added to the medium. For the anoxic cultures, 70-ml vials with 30 ml medium were flushed with pure nitrogen gas for 20 min, sealed with sterile septum caps and autoclaved. Culture bottles were incubated at 30 °C (unless stated otherwise) in the dark. In the determination of optimal growth conditions, the pH, temperature and NaCl concentrations were adjusted in the *Methylophaga* medium 1403. For $N_2O$ measurements, 720-mL bottles with 60-ml medium were used. For oxic cultures, cultures were performed in Erlenmeyer flasks and shaken at 150 rpm and 30 °C (*Mauffrey et al., 2017*). Bacterial growth was monitored by spectrophotometry ($OD_{600 nm}$). Bacterial flocs were dispersed with a Potter-Elvehjem homogenizer prior to measurements when needed.

## DNA extraction of the biofilm and quantitative PCR

The frozen samples of the biofilm taken from the denitrification system and of the acclimated biofilm taken after the 5th transfer were thawed, and washed 3 times with 0.5 mL the TEN buffer (50 mM Tris–HCl pH 8.0, 10 mM EDTA.2Na, NaCl 150 mM) and 5,000 g centrifugation for 2 min. The biomass was dispersed in 0.5 mL TEN with lysozyme (2.5 mg/mL) and incubated at 37 °C, 30 min. After adding sodium dodecyl sulfate and β-mercaptoethanol (final concentrations of 2% and 1%, respectively), the biomass was subjected to three cycles of freeze-thaw (10 min dry ice/ethanol bath, followed by 5 min at 65 °C), then treated with proteinase K (50 µg/mL final concentration) at 45 °C for 2 h, and finally with RNAse A (20 µg/mL final concentration) at 37 °C for 15 min. The DNA was purified by extractions with phenol/chloroform/isoamyl alcohol (25:24:1 v/v) and chloroform/isoamyl alcohol (49:1), and by precipitation with 0.25 volume of 10 M ammonium acetate and 2 volumes of ethanol 95% (*Sambrook & Russell, 2001*). DNA concentration was determined by the quantiFluor dsDNA system (Promega, Madison, USA). Quantitative PCR (qPCR) was performed with SYBR green in 20-µL volume with the Fast Start essential DNA Green Master (Roche Diagnostics, Laval, QC, Canada) containing 200 µM of primers (Table 1) and 50 ng biofilm DNA. The amplifications were performed at 95 °C for 10 min, followed by 40 cycles at 95 °C for 20 s, at the annealing temperature (Table 1) for 20 s, and at 72 °C for 15 s. After amplifications, specificity of the PCR products and the presence of primer dimers were verified by performing a melt curve by increasing the temperature from 65 °C to 95 °C by increments of 1 °C per step with a pause of 5 s for each step. Reactions were performed in a Rotor-Gene 6000 real-time rotary analyzer (Qiagen Inc. Toronto, ON, Canada). The amplification efficiencies for all primer pairs varied between 0.9 and 1.1. The copy number of each gene per ng of biofilm was calculated according to standard curves using dilutions of a gel-purified, PCR-amplified fragment of the corresponding genomic region of strain JAM1 (*tagH*) or of strain GP59

**Table 1** PCR primers used for qPCR assays.

| Gene target | Sequence (5′-3′) | Annealing temperature (°C) | Fragment length (nt) | Comment |
|---|---|---|---|---|
| *narG1* (strain JAM1/GP59; locus Q7A_446). Nitrate reductase subunit alpha (Nar1 system) | | | | |
| narG1 | AGCCCACATCGTATCAAGCA | 61 | 149 | qPCR |
| narG1 | CCACGCACCGCAGTATATTG | | | |
| *tagH* (strain JAM1, locus tag: Q7A_1110). Teichoic acid export ATP-binding protein[a] | | | | |
| TagH Forward | CCGTCATTTCGCTTCAAGAT | 55 | 711 | Standard |
| TagH Reverse | TCATGGCTTTTTCAGCCTTT | | | |
| qTagH Forward | GTTGCAAGGCTATAGTCGGAGT | 55 | 119 | qPCR |
| qTagH Reverse | TGGTACGCATTCCAGATGAATA | | | |
| *nirK* (strain GP59, locus tag: CDW43_15165). Copper containing nitrite reductase | | | | |
| NirK Forward | CGTTCAATACATGGGGTAAAGG | 55 | 1,134 | Standard |
| NirK Reverse | TGGGGCACAGTGATAAACAA | | | |
| qNirK Forward | AAGTCGGTAAAGTAGCCGTTGA | 55 | 138 | qPCR |
| qNirK Reverse | TCTCCATCGTCATTTGAACAAC | | | |
| *napA* (strain NL23, locus tag W911_13875) periplasmic nitrate reductase | | | | |
| qnapA Forward | AGGACGGGCGGATCAATTTT | 61 | 131 | qPCR |
| qnapA Reverse | CGGATATGCATCGGACACGA | | | |

**Notes.**
[a]The function was deduced by RAST annotations.
GenBank accession number of the genome of strain JAM1: CP003390.3, strain GP59: CP021973.1, and strain NL23: CP006912.1.

(*nirK*) (Table 1), or using dilution of the NL23 genomic DNA (*napA*) or the JAM1 genomic DNA (*narG1*) as the template.

## Genome sequencing and annotations

High molecular weight total DNA was extracted from GP59 and JAM1 pure cultures with the DNeasy Blood & Tissue Extraction Kit according to the manufacturer (Qiagen, Hilden, Allemagne). DNA samples were sent to the Genome Quebec Innovation Center for genome sequencing and assembly by using the PacBio technology. Genome assembly used was the HGAP workflow version April 2016 (*Chin et al., 2013*). Coverage was over 260 times with both genomes. Genome assembly was checked manually with the DNASTAR software (version 14.1) (Madison, WI, USA) for confirming or correcting pseudogenes. Gene annotations for the JAM1 genome were updated accordingly (Genbank accession number CP003390.3). For the GP59 genome, gene annotations were performed at NCBI (GenBank accession number CP021973). Annotations were also performed for both genomes at the Rapid Annotation using Subsystem Technology (RAST) web site (http://rast.nmpdr.org/seedviewer.cgi) and used for further analyses (*Aziz et al., 2008*; *Overbeek et al., 2014*).

Alignment between GP59 and JAM1 genomes was performed at RAST and with the Mauve Multiple Genome Alignment version 20150226 (progressive alignment) (*Darling et al., 2004*). The resulting alignments were corrected manually to precisely determine the common and unique genes, but also the intergenic sequences. Riboswitches and non coding RNA (ncRNA) sequences were determined by the NCBI annotation system. The average nucleotide identity (ANI) analysis was performed at http://enve-omics.ce.gatech.edu/ani/

with window size 1,000 bp, step size 200 bp, minimum identity 70%, minimum length 700 bp, minimum alignment 50 (*Goris et al., 2007*; *Rodriguez & Konstantinidis, 2016*). Genomes were analysed for tandem repeats at https://tandem.bu.edu/trf/trf.html (*Benson, 1999*).

## NirK Phylogenetic analysis

The deduced amino acid sequence of the GP59 *nirK* was compared with public protein databases by BLASTP at the NCBI web site to retrieve the closest NirK sequences. These sequences were than aligned with Cobalt (*Papadopoulos & Agarwala, 2007*). Evolutionary analysis was conducted using MEGA6.06 (*Tamura et al., 2013*) with the Maximum Likelihood method based on the Le_Gascuel_2008 model (*Le & Gascuel, 2008*). Initial tree(s) for the heuristic search were obtained automatically by applying Neighbor-Join and BioNJ algorithms to a matrix of pairwise distances estimated using a JTT model, and then selecting the topology with superior log likelihood value. A discrete Gamma distribution was used to model evolutionary rate differences among sites (five categories [+G, parameter = 0.8954]). The analysis involved 21 amino acid sequences. There were a total of 378 positions in the final dataset.

## The transcriptomes of *M. nitratireducenticrescens* GP59 and JAM1

Anoxic cultures of strains GP59 and JAM1 were performed in triplicate in the *Methylophaga* medium 1403 supplemented with 21.4 mM NaNO₃. Cells were harvested when $NO_3^-$ was almost reduced in all cultures. Total RNA extraction, RNA processing, cDNA sequencing and transcriptome analyses were described by *Mauffrey, Martineau & Villemur (2015)*, with the exception that we used Transcripts Per Million (TPM) for normalization instead of Reads Per Kilobase Million. One replicate culture of strain GP59 generated inconsistent results and was not considered in the transcriptome analyses. Intergenic regions were also considered in the transcriptome analyses to detect possible missed annotated genes or non-coding expressed sequences. An intergenic region includes the sequence from the stop codon of a gene to the start/stop codon of the adjacent gene. This sequence was then trimmed of 50 nucleotides at both ends to exclude as much as possible the transcript reads that would belong to the adjacent genes. Significance for difference in gene expression of the corresponding genes/sequences (defined as TPM) between strain JAM1 (triplicate cultures) and strain GP59 (duplicate cultures) was performed with the R Bioconductor NOIseq package v2.14.0 (NOIseqBio with replicates and $q = 0.9$) (*Tarazona et al., 2011*) and run with the R software v3.2.3 (*R Core Team, 2015*). Genes/sequences of one strain that had >2-fold higher level of TPM than the other strain showed significant differences. RNA-seq reads were deposited in the Sequence Read Archive (SRA) under the numbers SRP066381 (strain JAM1) and SRP132510 (strain GP59) at NCBI.

# RESULTS AND DISCUSSION

## Biofilm acclimation to batch cultures

The biofilm of the denitrification system (operated under continuous mode) was dispersed in vials containing 20 Bioflow carriers, the same used in the denitrification system, and cultured under anoxic, batch-mode conditions with $NO_3^-$. Only the carriers with the

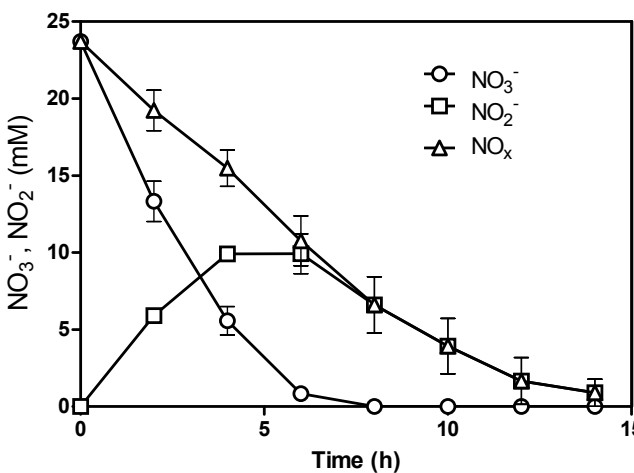

**Figure 1 $NO_3^-$ and $NO_2^-$ reduction by the acclimated biofilm.** The biofilm was batch-cultured under anoxic conditions. After the 5th transfer, the $NO_3^-$ and $NO_2^-$ concentrations were measured over different time intervals. $NO_x$: combined measurements of $NO_3^-$ and $NO_2^-$. Each point is the average with standard deviation of triplicate cultures.

growing biofilm were then transferred each week in fresh medium and cultured under the same conditions (see 'Material & Methods' Fig. S1). At the end of the 5th transfer culture when the $NO_3^-$ and $NO_2^-$ consumption rates stabilized, the buildup of biofilm was enough (Fig. S2) to extract total DNA and to perform culture assays on agar plates.

$NO_3^-$ and $NO_2^-$ concentrations were measured at regular interval times during the 5th transfer culture. Complete reduction of $NO_3^-$ and $NO_2^-$ ($NO_x$) was achieved within 15 h, at 1.8 mM $NO_x$ $h^{-1}$ (Fig. 1). This is 5-times faster than what was measured by Laurin et al. (24) with 60 Bioflow carriers taken directly from the denitrification reactor that operated under continuous-mode conditions (0.37 mM $NO_x$ $h^{-1}$ or 5.2 $NO_x$-N $L^{-1}$ $h^{-1}$). This result suggests that changes occurred in the microbial population of the batch biofilm cultures. qPCR assays were performed to determine the concentrations of *M. nitratireducenticrescens* JAM1 (*narG1*-targeted primers) and *H. nitrativorans* NL23 (*napA*-targeted primers) in the original and acclimated biofilm. In the original biofilm, strain JAM1 was at $2.5 \pm 0.6 \times 10^4$ *narG1*-copies/ng biofilm DNA and *H. nitrativorans* NL23 at $6.2 \pm 0.4 \times 10^5$ *napA*-copies/ng biofilm DNA. In the acclimated biofilm, the concentration of strain JAM1 increased by one order of magnitude ($3.8 \pm 0.9 \times 10^5$ *narG1*-copies/ng biofilm DNA), but decreased substantially for strain NL23 ($2.0 \pm 0.5 \times 10^2$ *napA*-copies/ng biofilm DNA). With the decrease of strain NL23 and the increase of strain JAM1 in the acclimated biofilm with a gain of denitrification performance, we hypothesized that a subpopulation of *M. nitratireducenticrescens* that can perform the complete denitrification pathway was enriched in the acclimated biofilm.

## Strain GP59 displaced *H. nitrativorans* NL23 and *M. nitratireducenticrescens* JAM1

The acclimated biofilm from the 5th transfer cultures was dispersed and cultured onto a *Methylophaga* specific medium, and several isolates were screened for $NO_3^-$ and $NO_2^-$ consumption (Fig. S1). One isolate, which we named strain GP59, was capable to consume $NO_3^-$ and $NO_2^-$, and cultures showed gas production, suggesting full denitrifying activities by this strain. The 16S rRNA gene sequence of strain GP59 has 100% identity to that of *M. nitratireducenticrescens* JAM1.

The concentration of strain GP59 was measured in the acclimated biofilm by qPCR to determine whether this strain is a major denitrifier. To distinguish strain GP59 from strain JAM1, primers targeted strain-specific sequences (*tagH* for strain JAM1 and *nirK* for strain GP59) were developed based on their respective genome (Table 1; see below). The GP59 cell concentrations went from undetected (below limit of PCR detection) in the original biofilm to high level in the acclimated biofilm at $2.2 \pm 0.35 \times 10^5$ *nirK*-copies/ng biofilm DNA. This is 2–3 orders of magnitude higher than that of strain JAM1 ($4.0 \pm 1.7 \times 10^2$ *tagH*-copies/ng biofilm DNA). These results confirm our hypothesis that a new *M. nitratireducenticrescens* strain with full denitrification capacity occurred in the acclimated biofilm, and displaced *H. nitrativorans* NL23, but also *M. nitratireducenticrescens* JAM1. Comparative analyses were then performed between strain GP59 and *M. nitratireducenticrescens* JAM1 at the physiological, genomic and transcriptomic levels.

## Physiological characterization of strain GP59

Strain GP59 cultured under anoxic conditions (planktonic pure cultures) showed a 24-h lag before growth occurred (Fig. 2A). The growth yields reached about 1.2 $OD_{600nm}$ with 42.8 mM $NO_3^-$ exposure and did not increase significantly in cultures exposed with higher $NO_3^-$ concentrations. Strain JAM1 cultured under the same conditions showed no lag phase, and growth yields were 3- to 6-fold lower than those of the GP59 cultures (Fig. 2A). The maximum specific growth rate ($\mu$max) and the half-saturation constants of $NO_3^-$ for growth (Ks) were calculated (Table 2). The $\mu$max for strain GP59 is higher (3.3 times) than that of strain JAM1. To assess the affinity of strain GP59 toward $NO_3^-$ for growth, the $\mu$max/Ks ratio was calculated (*Healey, 1980*) (Table 2). This ratio (1.2 $\mu M^{-1}$ $NO_3^-$ $h^{-1}$) is not different than the one calculated for strain JAM1 at 1.3 $\mu M^{-1}$ $NO_3^-$ $h^{-1}$ (*Mauffrey, Martineau & Villemur, 2015*). These results concur with the genome sequences (see below) with near 100% identity between the two strains in gene clusters encoding the two Nar systems and the $NO_3^-$ transporters (NarK).

As observed with growth, a 24-h lag period was observed in the GP59 cultures before $NO_3^-$ started to be consumed, whereas complete $NO_3^-$ consumption occurred in the JAM1 cultures within 24 h (Fig. 2B). The $NO_3^-$ reduction rates increased linearly with the increase of $NO_3^-$ concentrations in the GP59 cultures (Fig. 3A). The specific $NO_3^-$ reduction rates (rates normalized by the biomass) averaged around 1.5 to 2 mM $NO_3^-$ $h^{-1}$ $OD^{-1}$ and showed no significant changes at any $NO_3^-$ concentrations tested (Fig. 3B). In the JAM1 cultures, the $NO_3^-$ reduction rates reached a plateau at 24 mM $NO_3^-$ (Fig. 3A), and, as observed with the GP59 cultures, showed no significant changes in the specific $NO_3^-$

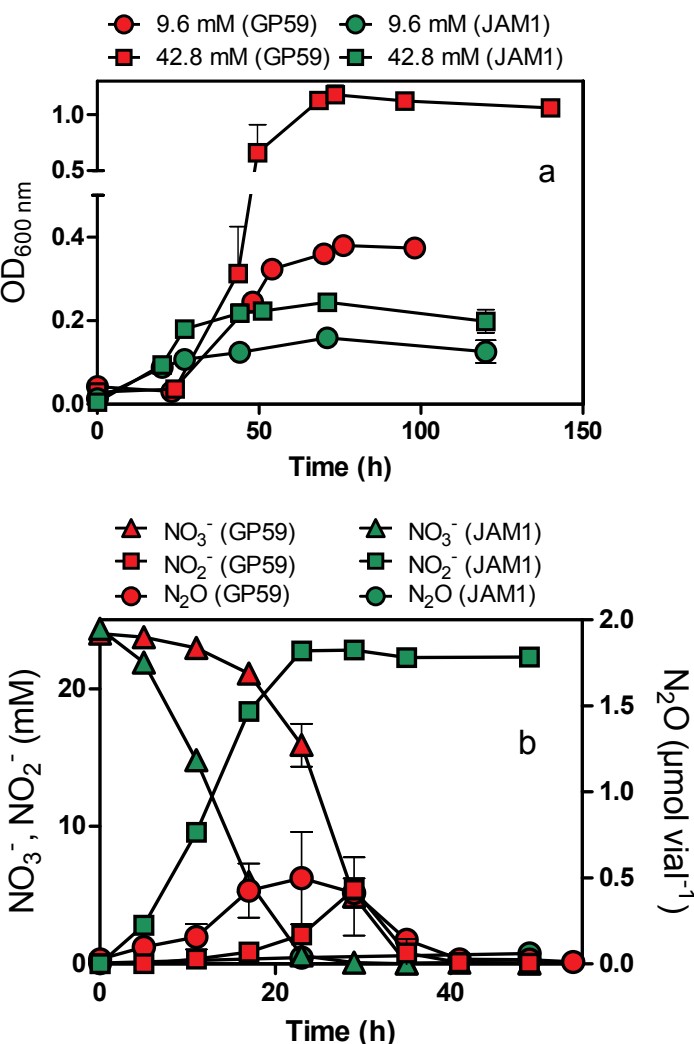

**Figure 2  Growth, and NO$_3^-$, NO$_2^-$ and N$_2$O dynamics by *Methylophaga nitratireducenticrescens* GP59 and JAM1.** (A) Growth. Strain GP59 and strain JAM1 were cultured under anoxic conditions with 9.6 or 42.8 mM NO$_3^-$. (B) NO$_3^-$, NO$_2^-$ and N$_2$O dynamics. Strain GP59 and strain JAM1 were cultured under anoxic conditions with 21.4 mM NO$_3^-$, and NO$_3^-$, NO$_2^-$ and N$_2$O were measured over different time intervals. Each point is the average with standard deviation of triplicate cultures for (A), and of duplicate cultures for (B).

reduction rates (averaged around 4 to 6 mM NO$_3^-$ h$^{-1}$ OD$^{-1}$) (Fig. 3B). Interestingly, these specific rates are 2.4 to 5.4-fold higher than those of strain GP59, suggesting that the JAM1 cells have a higher dynamism of NO$_3^-$ processing (e.g., NO$_3^-$ intake and reduction) than the GP59 cells. Close examination of the respective genomes and transcriptomes (see below) did not reveal specific gene(s) that would explain these differences.

N$_2$O was detected in the GP59 anoxic cultures and reached maximum accumulation (*ca.* 0.5 μmol N$_2$O vial$^{-1}$ or 0.04% N-input) when NO$_2^-$ peaked in the medium, and then decreased in concentration afterwards (Fig. 2B). This result concurs with the presence of a

**Table 2  Kinetics of growth under anoxic conditions.**

|  | Strain GP59 | Strain JAM1[a] |
|---|---|---|
| $\mu$max ($h^{-1}$) | 0.0380 (0.0027) | 0.0116 (0.0008) |
| Ks (mM) | 30.7 (5.8) | 9.2 (1.9) |
| $\mu$max/Ks ($\mu M^{-1}\,h^{-1}$) | 1.2 (0.2) | 1.3 (0.3) |

**Notes.**

$\mu$max, maximum growth rates. Ks, half-saturation constants of $NO_3^-$ for growth. Values between parentheses are standard deviation of triplicates.

[a] From *Mauffrey, Martineau & Villemur (2015)*.

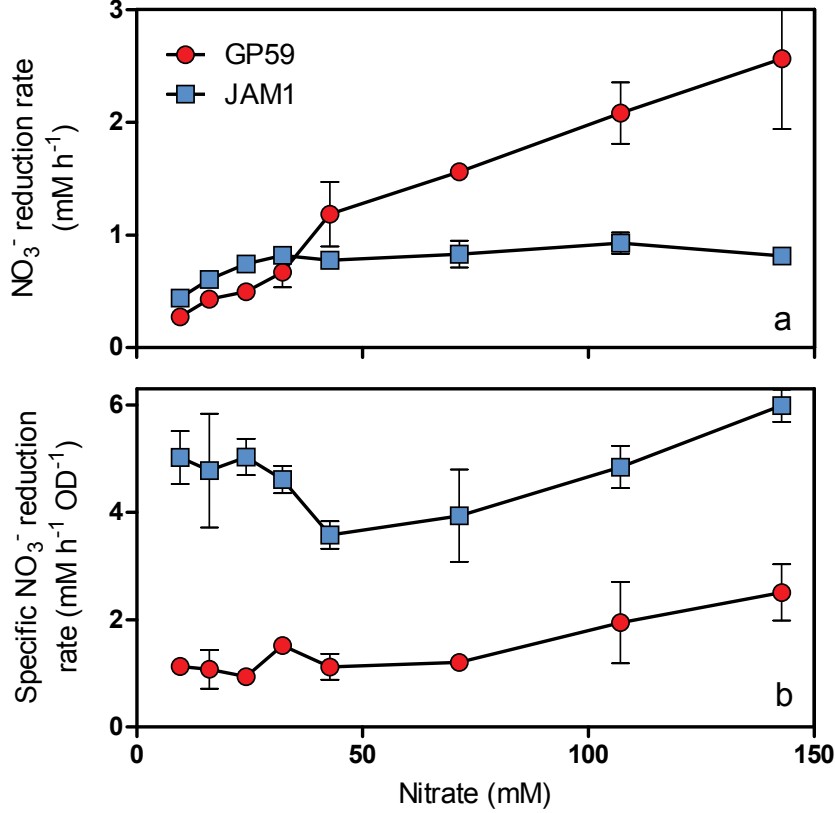

**Figure 3  Specific $NO_3^-$ reduction rates.** (A) $NO_3^-$ reduction rates of strains GP59 and JAM1. Each point is the average with standard deviation of triplicate anoxic cultures. (B) Specific $NO_3^-$ reduction rates. These rates were calculated with the $NO_3^-$ reduction rates by the generated culture biomass ($OD_{600nm}$) at the end of the exponential phase.

complete denitrification pathway in strain GP59 as revealed by its genome sequence (see below).

Strain GP59 was cultured under oxic conditions with different concentrations of $NO_3^-$ (9.6 to 142.8 mM). Contrary to the JAM1 cultures that can reduce $NO_3^-$ to $NO_2^-$ under oxic conditions (*Mauffrey, Martineau & Villemur, 2015*), no $NO_3^-$ consumption was observed in the GP59 cultures at any tested concentrations. These latter results, coupled with the

24-h lag period before nitrate reduction occurs in the GP59 anoxic cultures, suggest that strains GP59 and JAM1 regulate differently the expression of their denitrification genes.

Both strains cannot grow on methylamine and fructose. They showed similar growth profile related to NaCl concentrations (optimal growth at 1 to 5% NaCl; weaker growth at 8%; no growth at 0 and 10%), pH (optimal growth at pH 8; weaker growth at pH 7, 9 and 10; no growth at pH 6 and 11) and temperature (optimal growth at 30 °C; variability in culture replicates at 22 and 34 °C; weak growth at 37 °C). Sensibility to antibiotics by strain GP59 was also similar to those reported by *Villeneuve et al. (2013)* for strain JAM1 (chloramphenicol, trimethoprim, ampicillin, tetracycline, streptomycin, gentamycin and kanamycin).

## The genome of strain GP59 contains nirK

The genome of strain GP59 was sequenced by the PacBio technology. Although strain JAM1 was already sequenced by the pyrosequencing technology, it was resequenced by the PacBio technology to correct some discrepancies in the genome assembly (Data S1). Analysis of the GP59 genome revealed the presence of two prophages (named GPMu1 and GPMu2) integrated side by side in the same orientation and separated by 11,102 nt (Data S2). The intercalated sequence has a GC content of 40.1%, which is lower than in the overall genome (44.8%). Gene arrangement of the prophages resembles to that of the *E. coli* phage Mu and of the *Haemophilus influenzae* FluMu prophage (*Morgan et al., 2002*). Gene clusters encoding Mu-type prophage proteins are also present in the genome of two *Methylophaga* species such as *M. frappieri* JAM7 that was also isolated from the original biofilm of the denitrification system (*Auclair et al., 2010*) (Data S2). No prophage was found in the JAM1 genome.

The 11,102 nt intercalated sequence contains a gene encoding a NirK-type $NO_2^-$ reductase of 363 amino acid residues (Data S2). Upstream of this *nirK*, is a gene encoding the NO reductase activation protein NorD. The precise function of NorD is yet to be known, but the *norD* mutant in *Paracoccus denitrificans* showed no activity of the $cd_1$-type $NO_2^-$ reductase (NirS) and of the NO reductase (*DeBoer et al., 1996*). Available genomic sequences from other *Methylophaga* species and from metagenomic studies were screened for the presence of genes encoding NirK highly similar to the GP59 NirK. The closest affiliations of the GP59 NirK are with NirK encoded in three reconstituted *Methylophaga* sp. genomes from a metagenomic study of marine subsurface aquifer samples (NCBI Bioproject PRJNA391950 by Tully, Wheat, Glazer and Huber, University of Southern California) (Fig. 4). These three NirK have 84–85% similarity with the GP59 NirK sequence. In addition, *norD* is also present adjacent to *nirK* in these three *Methylophaga* sp. genomes.

The previous results suggest that there are *Methylophaga* strains in natural environments that carry denitrification modules. We further performed data mining to retrieve denitrification genes in available genomes of *Methylophaga* species or in reconstituted *Methylophaga* genomes from metagenomic studies. Data provided by the Bioproject PRJNA391950 revealed that the metagenome of the *Methylophaga* sp. isolate NORP53 (accession number NVVW00000000) contains two gene clusters with one (contig NVVW01000003) having the same gene arrangement of the Nar1 gene cluster in the

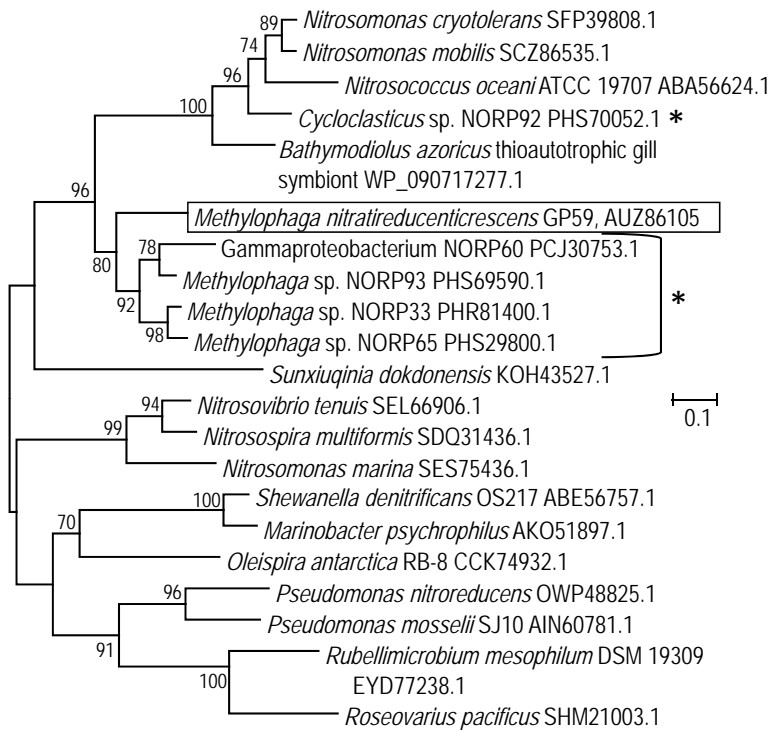

**Figure 4 Phylogenetic analysis of NirK of *Methylophaga nitratireducenticrescens* GP59.** An unrooted phylogenetic tree demonstrating the evolutionary relationship of *nirK* deduced amino acid sequences is illustrated. The evolutionary history was inferred by using the Maximum Likelihood and the tree with the highest log likelihood (−7,598.4857) is shown. The percentage (values above 70%) of trees in which the associated taxa clustered together is shown next to the branches (bootstrap analysis with 1,000 replicates). The tree is drawn to scale, with branch lengths measured in the number of substitutions per site. The GenBank accession number is provided after each bacterial name. *: The *Cycloclasticus* sp. NORP92, Gammaproteobacterium NORP60, and the *Methylophaga* sp. NORP93, NORP33 and NORP65 are from the NCBI bioproject PRJNA391950 by Tully, Wheat, Glazer and Huber, University of Southern California.

JAM1 and GP59 genomes (*narXL, NarK1K2GHJI*), and another one (NVVW01000010) encoding a N$_2$O reductase (*nosRZDFYL*). At the amino acid sequence level, these gene clusters of isolate NORP53 are 67 to 90% similar with the corresponding gene products in the JAM1 and GP59 genomes. Finally, *nirK* sequences were found in the genome of *M. frappieri* JAM7 and of 34 *Methylophaga* sp. retrieved from metagenomic studies (Data S3). Although *M. frappieri* JAM7 does not grow under anoxic conditions with NO$_3^-$, which concurs with the absence gene cluster encoding a dissimilatory NO$_3^-$ reductase, the occurrence of *nirK* may be associated with the detoxification mechanism. All these results suggest that the different denitrification modules are carried by some *Methylophaga* species present in natural environments.

## Strain GP59 belongs to the species Methylophaga nitratireducenticrescens

Table 3 summaries the overall features of the GP59 and JAM1 genomes. Both genomes share 2,790 coding sequences (CDS) and 11 riboswitches (Data S4). The 67 kbp chromosomic

**Table 3  Genomic features of *M. nitratireducenticrescens* strains JAM1 and GP59.**

| | JAM1 | GP59 | |
|---|---|---|---|
| Genome | | | |
| Length (nt) | 3,137,100 | 3,238,484 | |
| GC content (%) | 44.75 | 44.86 | |
| CDS[a] | 3,027 | 3,187[b] | |
| Common CDS | 2,790 | 2,790 | |
| tRNA | 44 | 44 | |
| rRNA (5S-16S-23S) | 9 | 9 | |
| ncRNA | 4 | 4 | |
| (TmRNA, SRP-RNA, RNase P, 6S RNA) | | | |
| Riboswitch | 11 | 11 | |
| Small Tandem Repeats[c] | | | |
| TCAGYCA | 16 (785,393) | 12 (710,616) | |
| CTTCGG | 55 (2,294,123) | 19 (2,287,110) | |
| GGYTCT | 39 (2,660,735) | 37 (2,669,276) | |
| Large tandem repeat[d] | | | |
| 3095-nt repeats | 1 (1,940,231; 93%[d]) | 4 (1,904,086; 100%[d]) | |
| Plasmids | None | pGP32 | pGP34 |
| Length (nt) | | 32,421 | 33,560 |
| GC content (%) | | 44.29 | 44.13 |
| CDS | | 51 | 52 |

**Notes.**
[a] Based on RAST annotations.
[b] The possible CDS in the ambiguous region of GPMu1 were not taking into account. Common CDS: see Data S4.
[c] Number of copies. Genome location under parenthesis. Only repeats with more than 10 copies are presented.
[d] Number of copies. Genome location under parenthesis. % identity between the 3095-nt long repeats (see text).

region containing the denitrification gene clusters *nar1, nar2, nor1, nor2* and *nos* (*Mauffrey, Martineau & Villemur, 2015*) is 99.94% identical (43 substitutions, no gap) between the two strains. Based on the genome annotations, functions associated with CDS that are unique to strain JAM1 have in most cases an equivalent in strain GP59. In addition to CDS associated with the two prophages and the intercalated sequence, the GP59 genome has CDS associated with an integrated plasmid (position 724937 to 765476; Data S4). The GC skew between the two genomes is relatively similar (Fig. S3). Two-way ANI analysis showed 99.30% identity, and >80% conserved sequences between the two genomes, indicating that both strains belong to the same species (*Goris et al., 2007*; *Rodriguez & Konstantinidis, 2016*).

Both genomes have small tandem repeats with >10 repeats positioned at equivalent locations. These tandem repeats vary in number of repeats between the two strains (Table 3), and are located in CDS with no putative known function. A long tandem repeat (four 3095-nt repeats) was found in the GP59 genome (Table 3). Genes encoding an aspartate aminotransferase and a TonB-dependent siderophore receptor are among the three genes repeated 4 times. The equivalent region in the JAM1 genome shows no repetition, with 93% identity with the 3095-nt repeat sequence with most of the divergence in the TonB-dependent siderophore receptor gene.

Strain GP59 contains two plasmids here named pGP32 and pGP34 (Table 3). This result concurs with the observation of extra-chromosomic DNA when plasmid DNA extraction was performed on GP59 biomass (Data S5). These plasmids encode type IV secretion system as observed in many plasmids hosted by Gram negative bacteria (Data S5) (*Christie, 2001*). They also carry genes encoding for recombinases, nucleases, DNA methyltransferases, toxin/antitoxin, and carbon storage regulator. These two plasmids show no homology with the integrated plasmid in the GP59 genome. *M. frappieri* JAM7 also carries a 48.5 kb plasmid, which resembles in its structure to the IncPalpha plasmids (Data S5). It shares no homology with the pGP32 and pGP34 plasmids.

### *M. nitratireducenticrescens* GP59 did not evolve from *M. nitratireducenticrescens* JAM1

Did strain GP59 evolve during the acclimation process of the biofilm from strain JAM1 that underwent series of chromosomic rearrangements and phage infections, or was strain GP59 present in the original biofilm and enriched during the acclimation? To try to answer this question, we analyzed the Clustered Regularly Interspaced Short Palindromic Repeats (CRISPR) region that has been identified in the GP59 and JAM1 genomes. The CRISPR system is an adaptive immunity system that is present in most archaea and many bacteria and that acts against invading genetic elements, such as bacteriophages and plasmids. The chromosomic arrangement of CRISPR system includes CRISPR associated genes (*cas*), a leader sequence upstream of an array of short repeats interspersed with unique spacers that are almost identical to fragments of bacteriophage and plasmid genes. The transcription of the spacer array provides complementary DNA targeting, for instance, phage genome resulting in target DNA degradation (*Barrangou & Marraffini, 2014*; *Garneau et al., 2010*; *Makarova et al., 2011*). Addition of new spacers in response, for instance, of bacteriophage infection occurs proximal to the leader sequence. Accordingly, CRISPR spacers provide a historical perspective of phage exposure, with spacers at the vicinity of the leader were relatively recently added, and those distal spacers likely originated from previous events (*Horvath et al., 2008*).

The CRISPR region in GP59 and JAM1 genomes comprises 5 associated CRISPR genes (Fig. 5A). Except for the spacer sequences, the nucleotide sequence of the CRISPR genes and the repeat sequences are identical in both genomes (Figs. 5A, 5C). There are 105 and 115 spacers in the JAM1 and GP59 CRISPR regions, respectively (Fig. 5B). In the JAM1 CRISPR region, 14 spacer sequences were found twice. The GP59 and JAM1 CRISPR regions share 29 spacer sequences (Fig. 5B); all of them are distal to the leader sequence (the most ancient acquired sequences, *Horvath et al., 2008*). No spacer sequence was found twice in the GP59 CRISPR region. As both CRISPR regions do not share sequence between their proximal spacers, strains GP59 and JAM1 underwent a different history of phage infections from a common ancestry. This strongly suggests that strain GP59 did not originate from strain JAM1 that underwent series of chromosomic rearrangements and phage infections in a short period of time and in a closed environment, but rather was present in the original biofilm and was enriched during the acclimation phase of the biofilm.

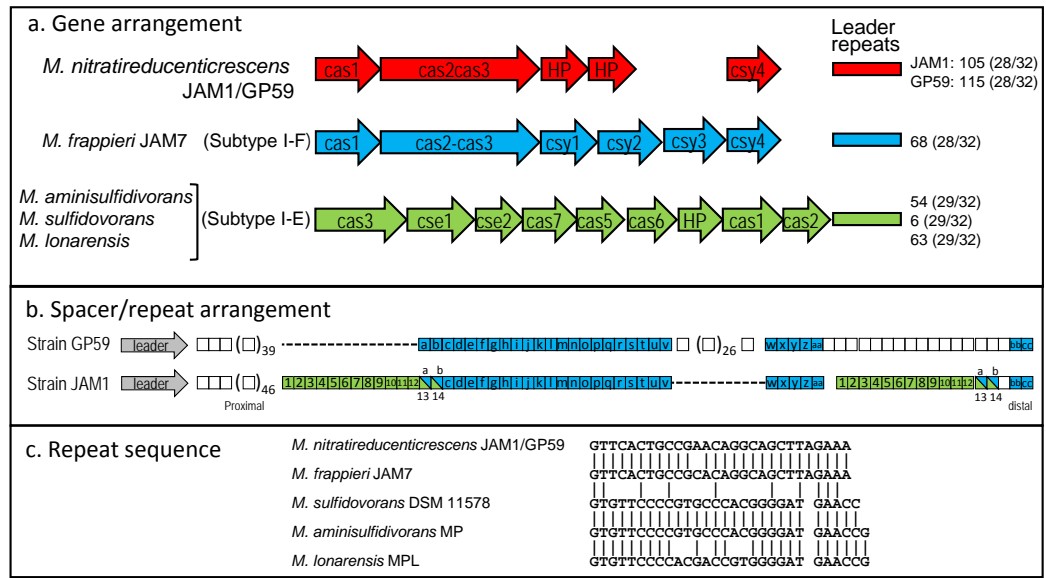

**Figure 5 CRISPR chromosomal arrangement in *Methylophaga* genomes.** (A) Gene arrangement. Sequences were retrieved from GenBank. Accession numbers: *M. nitratireducenticrescens* JAM1: CP003390 (Q7A_2613 to Q7a_2609) and GP59: CP021973 (CDW43_11545 to CDW43_11525); *M. frappieri* JAM7 CP003380 (Q7C_1045 to Q7C_1040); *M. aminisulfidivorans* MP: NZ_AFIG01000001 (MAMP_RS08465 to MAMP_RS08425), *M. sulfidovorans* DSM 11578: FOSH01000003 (SAMN04488079_1032 to SAMN04488079_10310); *M. lonarensis* MPL, APHR01000054 (MPL1_10022 to MPL1_10062). Nomenclature is based on *Barrangou & Marraffini (2014)*. HP, Hypothetical protein. Numbers at the right are the number of repeat units with the repeat/spacer nucleotide length in parentheses. (B) Spacer/repeat unit arrangement. Blue: the 29 common units. Green: the 14 units repeated twice in strain JAM1. (C) CRISPR repeat sequences in different *Methylophaga* species.

The GP59 and JAM1 CRISPR regions were also compared to other available *Methylophaga* genomes. *M. frappieri* JAM7 contains 68 repeat units. The repeat sequence has one-nt substitution with the GP59/JAM1 one (Fig. 5C). None of its spacer sequences were found in the GP59 and JAM1 CRISPR regions. In the *M. aminisulfidivorans* MP, *M. sulfidovorans* DSM 11578, *M. lonarensis* MPL genomes, 59, 6 and 63 spacer sequences were found, respectively; none were repeated twice or were found in other *Methylophaga* species.

The CRISPR genes and gene arrangement in *M. frappieri* JAM7 are typical of the subtype I-F CRISPR-Cas system, whereas those of *M. aminisulfidivorans* MP, *M. sulfidovorans* DSM 11578, *M. lonarensis* MPL are related to the subtype I-E one (Fig. 5A) (*Makarova et al., 2011*). In strains GP59 and JAM1, the CRISPR genes and gene arrangement cannot be associated to a defined subtype. However, the gene arrangement and the deduced amino acid sequences of *cas3* and of the two downstream CDS are highly similar to CRISPR genes found in several bacterial genomes such as *Legionella brunensis* (GenBank accession number LNXV01000004) and *Moraxella atlantae* (GenBank accession number LZMZ01000013).

## Transcriptome analyses

The GP59 and JAM1 genomes share more than 85% of their genes. However, gene variations, the presence of strain-specific genes, and the occurrence of plasmids in strain

**Table 4  Transcript levels of genes involved in the nitrogen pathway in *Methylophaga nitratireducenticrescens* JAM1 and GP59.**

| Locus Tag | Gene/function | [a]Expression ratio JAM1/GP59 | [b]*mxaI* = 100 | |
|---|---|---|---|---|
| | | | GP59 | JAM1 |
| **Nitric oxide reductase 1** | | | | |
| CDW43_RS01925-40 | *norCBDQ* | 0.93 | 11.7 | 12.4 |
| CDW43_RS01945 | *norR* | 1.42 | 4.9 | 6.8 |
| CDW43_RS01950 | *norE* | 0.84 | 7.8 | 6.5 |
| **Nitrate reductase 1 (with transporters and regulators)** | | | | |
| CDW43_RS01975-80 | *narXL* | 2.07 | 3.8 | 7.9 |
| CDW43_RS01985-2010 | *narK1K2narGHJI* | 0.66 | 17.5 | 9.8 |
| **Nitrous oxide reductase** | | | | |
| CDW43_RS02060-85 | *nosRZDFYL* | 1.51 | 5.5 | 9.0 |
| **Nitrate reductase 2 (with transporter)** | | | | |
| CDW43_RS02155 | *narK* | 0.75 | 1.9 | 1.4 |
| CDW43_RS02165-80 | *narGHJI* | 0.83 | 1.9 | 1.5 |
| **Nitric oxide reductase 2** | | | | |
| CDW43_RS02185-2200 | *norCBQD* | 0.83 | 2.1 | 1.7 |
| **Copper containing nitrite reductase** | | | | |
| CDW43_RS15160 | *nirK* | na | 1.0 | na |
| **Nitrate assimilatory pathway** | | | | |
| CDW43_RS11575 | Nitrate/nitrite transporter | 1.17 | 0.2 | 0.2 |
| CDW43_RS11585 | Assimilatory nitrate reductase large sub. | 1.17 | 0.3 | 0.4 |
| CDW43_RS11590 | Nitrite reductase [NAD(P)H] small sub. | 0.81 | 0.2 | 0.2 |
| CDW43_RS11595 | Nitrite reductase [NAD(P)H] large sub. | 0.84 | 0.4 | 0.3 |
| **Other genes involved in nitrogen pathway** | | | | |
| CDW43_RS01710 | *fnr* transcriptional regulator | 1.41 | 3.9 | 5.5 |
| CDW43_RS00310 | *nnrS* protein involved in response to NO | 2.71 | 4.4 | 12.2 |
| CDW43_RS01960 | *nnrS* | 0.99 | 7.4 | 7.3 |
| CDW43_RS08605 | *nnrS* | 1.61 | 3.8 | 6.4 |
| CDW43_RS00315 | *nsrR* transcriptional regulator | 1.90 | 17.4 | 33.8 |
| CDW43_RS01825 | *nsrR* | 0.86 | 2.3 | 2.0 |
| CDW43_RS00320 | *dnrN/norA/yftE* NO-dependent regulator | 1.98 | 25.0 | 48.7 |
| CDW43_RS01015 | Ammonium transporter | 1.01 | 1.0 | 1.0 |
| CDW43_RS05605 | Ammonium transporter | 0.48 | 0.9 | 0.5 |
| CDW43_RS01835 | Nitric oxide dioxygenase | 0.79 | 2.7 | 2.2 |
| CDW43_RS09635 | Nitric oxide dioxygenase | 0.65 | 8.0 | 5.0 |
| CDW43_RS01010 | Nitrogen regulatory protein P-II | 0.85 | 1.2 | 1.0 |
| CDW43_RS06615 | Nitrogen regulatory protein P-II | 0.80 | 7.0 | 5.5 |
| CDW43_RS13530 | Nitrogen regulatory protein P-II | 0.74 | 1.2 | 0.8 |
| CDW43_RS01095 | Nitrogen regulation protein NtrY | 1.30 | 1.4 | 1.9 |
| CDW43_RS01100 | Nitrogen regulation protein NtrX | 1.35 | 3.6 | 4.9 |
| CDW43_RS06315 | Nitrogen regulation protein NtrB | 0.60 | 1.1 | 0.6 |

**Table 4** (*continued*)

| Locus Tag | Gene/function | [a]Expression ratio JAM1/GP59 | [b]*mxaI* = 100 GP59 | [b]*mxaI* = 100 JAM1 |
|---|---|---|---|---|
| CDW43_RS06320 | Nitrogen regulation protein NR(I) | 1.12 | 0.7 | 0.8 |
| CDW43_RS15625 | PTS IIA-like nitrogen-regulator PtsN | 0.95 | 12.1 | 11.1 |

**Notes.**
[a]Ratio of JAM1 transcripts per million (TPM) by GP59 TPM.
[b]Transcript levels relative to the expression of the methanol dehydrogenase small unit (*mxaI*) set to 100 (see Data S6). For polycistronique operons, the values are the average values of individual genes.
na, not applicable.
Locus tag refers to the genome sequence of strain GP59 (GenBank accession number CP021973). Gene/function is based on RAST annotations, see Data S4.

GP59 could affect the expression pattern of these common genes. To assess whether these genes were expressed differently between strains GP59 and JAM1, the transcriptome of both strains was derived from anoxic cultures at the end of the exponential growth when $NO_3^-$ was nearly completely reduced in both cultures. Among the 2857 common genes (CDS and ncRNA), more than 85% of those in strain GP59 showed no significant difference in the transcription levels (<2-fold difference in TPM) (Data S4) with their counterparts in strain JAM1. The vast majority of genes involved in the nitrogen pathways were expressed in equivalent levels in both strains (Table 4). The fact that the cultures were sampled for RNA extraction when both strains were at an equivalent physiological phase (anoxic conditions with $NO_3^-$ almost consumed) could explain this high proportion of similar gene expression between the two strains. Among genes with significant differences are those encoding the synthesis of the osmoprotectant ectoine, which were 3- to 6-fold higher in the JAM1 cultures than in the GP59 cultures (Fig. S4). This suggests that strain JAM1 could be more resilient to changes in salinity.

The most expressed genes were sorted out from both transcriptomes to assess which specific metabolic pathways are the most solicited under denitrifying conditions. High levels of gene expression were found in both strains for genes involved in growth, energy and respiration (e.g., ribosomal proteins, ATP synthase, pseudoazurin), and the C-1 metabolic carbon such as methanol dehydrogenase, ribulose monophosphate enzymes (D-arabino-3-hexulose 6-phosphate formaldehyde-lyase, 6-phospho-3-hexuloisomerase), formaldehyde activating enzymes (Data S6, Fig. S4). One of the highest expressed genes in both genomes is located in the chromosomic region comprising three adjacent genes encoding similar proteins (>88% similarity). Each of these genes has a cyclic di-GMP riboswitch in their respective upstream sequence. The first of the three genes showed high level of TPM in the riboswitch and in the coding sequence (Table 5). The other two genes and their respective riboswitch showed much lower expression levels. The deduced amino acid sequence of these three similar proteins contains a *carbohydrate-binding module 6* that is found in carbohydrate-active enzymes such as glycoside hydrolases (*Henshaw et al., 2004*), and three transmembrane domains. Upstream and downstream of these three genes are genes encoding proteins involved in polysaccharide export and in the capsular polysaccharide biosynthesis. These proteins may participate in this synthesis. Finally, the pGP32 plasmid accounted for 23.1% of total TPM in the GP59 transcriptomes, whereas it was 0.7% for the pGP34 plasmid (Data S6). In the pGP32 plasmid, nine genes accounted

**Table 5 Relative expression of selected genes under anoxic culture conditions.**

| Locus tag | Function | Strand | Relative expression | |
|---|---|---|---|---|
| | | | GP59 | JAM1 |
| CDW43_06820 | Mannose-1-phosphate guanylyltransferase | + | 2.3 | 3.5 |
| CDW43_06825 | Capsular polysaccharide synthesis enzyme CpsA, sugar transferase, undecaprenyl-phosphate glucose phosphotransferase | + | 3.3 | 5.2 |
| DW43_06835 | Cyclic di-GMP riboswitch class I<br>Hypothetical protein, carbohydrate-binding module 6 | + | 2,011<br>936 | 1,561<br>777 |
| CDW43_06840 | Cyclic di-GMP riboswitch class I<br>Hypothetical protein, carbohydrate-binding module 6 | + | 9.1<br>7.3 | 13<br>7.6 |
| CDW43_06845 | Cyclic di-GMP riboswitch class I<br>Hypothetical protein, carbohydrate-binding module 6 | + | 6.5<br>30 | 2.9<br>7.1 |
| CDW43_06850 | Periplasmic protein involved in polysaccharide export | + | 6.8 | 4.6 |
| CDW43_06855 | Hypothetical protein | + | 4.7 | 3.6 |
| CDW43_06860 | Exopolysaccharide transport protein, putative | + | 2.6 | 2.0 |

**Notes.**

Gene expression is relative to *mxaI* set to 100 (2602 TPM for strain GP59 and 4060 TPM for strain JAM1). Locus tag: from GenBank accession number CP021973. Functions are based on RAST annotations.

for 20% of total TPM, among which are genes involved in restriction-modification systems (Data S6).

## CONCLUSIONS

Culturing the denitrifying biofilm under batch conditions has favored the enrichment of a new denitrifying subpopulation, representing by *M. nitratireducenticrescens* GP59 that displaced *H. nitrativorans* NL23 and *M. nitratireducenticrescens* JAM1. In addition of complete denitrification by strain GP59, two important differences were observed between the two strains. Strain GP59 cannot reduce $NO_3^-$ under oxic conditions as does strain JAM1, and a 24-h lag period was observed in GP59 anoxic cultures before growth and $NO_3^-$ reduction occurs. These results suggest differences in regulation of the denitrification genes between the two strains. The genome of strain GP59 showed insertions of two Mu-type prophages that probably brought a *nirK* gene, the missing denitrification gene in strain JAM1. Both genomes share more than 85% of their genes, and these genes were expressed at similar level under anoxic conditions at the end of the exponential growth. Finally, analysis of the CRISPR region suggests that strain GP59 did not originate from strain JAM1, but was present in the original biofilm and was enriched during the acclimation process.

These comparative analyses between strains GP59 and JAM1 unveiled that *M. nitratireducenticrescens* species enriched in the biofilm of the bioreactor encompasses a mosaic population structure. Although plasticity of the genomic landscape of bacterial species resulting to gene rearrangements, point mutations and lateral gene acquisition was observed in nature (*Allen et al., 2007*; *Bendall et al., 2016*), the observation of *M. nitratireducenticrescens* GP59 reported here shows functional consequences of population variation on process rate and system performances. The reason why strain GP59 did occur in the acclimated biofilm but not in the Biodome denitrification system remains

obscure and is under investigation. This could be related to the batch-mode conditions where the transient accumulation of $NO_2^-$ (up to 10 mM; Fig. 1) could have adverse the growth of *H. nitrativorans* NL23, favoring the enrichment of *M. nitratireducenticrescens* GP59. In the denitrification reactor at the Biodome, such accumulation of $NO_2^-$ was not encountered probably due the continuous-operating mode or the lower denitrification rates that prevailed.

Our study broadens the ecology of *M. nitratireducenticrescens* with the occurrence of a microbial seedbank suited for anoxic conditions. So far among the genus *Methylophaga*, *M. nitratireducenticrescens* is the only reported species that can grow under anoxic conditions. Based on metagenomic studies, however, where denitrification genes were found in *Methylophaga*-reconstituted genomes, this anoxic metabolism within this genus is more widespread than previously thought. Methanol is generated by anthropogenic activities or by biogenic metabolisms such as plant growth and decomposition, and ocean ecosystems are thought to participate actively in its production. Furthermore, oceans are known to be a sink of atmospheric methanol (*Galbally & Kirstine, 2002*; *Millet et al., 2008*). As they are ubiquitous in marine environments, *Methylophaga* spp. are therefore important actors in the methanol turnover on Earth, either under oxic environments or, based on our studies with *M. nitratireducenticrescens*, under anoxic marine environments.

## ACKNOWLEDGEMENTS

We thank Karla Vasquez for her technical assistance.

### Funding

This research was supported by a grant to Richard Villemur from the Natural Sciences and Engineering Research Council of Canada # RGPIN-2016-06061. The funders had no role in study design, data collection and analysis, decision to publish, or preparation of the manuscript.

### Grant Disclosures

The following grant information was disclosed by the authors:
Natural Sciences and Engineering Research Council of Canada: #RGPIN-2016-06061.

### Competing Interests

The authors declare there are no competing interests.

### Author Contributions

- Valérie Geoffroy and Geneviève Payette conceived and designed the experiments, performed the experiments, analyzed the data, prepared figures and/or tables, authored or reviewed drafts of the paper, approved the final draft.
- Florian Mauffrey conceived and designed the experiments, performed the experiments, analyzed the data, authored or reviewed drafts of the paper.

- Livie Lestin performed the experiments, analyzed the data, approved the final draft.
- Philippe Constant conceived and designed the experiments, analyzed the data, contributed reagents/materials/analysis tools, authored or reviewed drafts of the paper.
- Richard Villemur conceived and designed the experiments, performed the experiments, analyzed the data, contributed reagents/materials/analysis tools, prepared figures and/or tables, authored or reviewed drafts of the paper, approved the final draft.

## DNA Deposition

The following information was supplied regarding the deposition of DNA sequences:

The genome of strain JAM1 is accessible via GenBank accession number CP003390.3. The genome of strain GP59 is accessible via GenBank accession number CP021973. The RNA-seq reads were deposited in the Sequence Read Archive (SRA) under the numbers SRP066381 (strain JAM1) and SRP132510 (strain GP59) in GenBank.

## Data Availability

The raw data and supplemental documents are available as Supplemental Files.

## Supplemental Information

Supplemental information for this article can be found online at http://dx.doi.org/10.7717/peerj.4679#supplemental-information.

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
