# Peer review of "Strain-level genetic diversity of Methylophaga nitratireducenticrescens confers plasticity to denitrification capacity in a methylotrophic marine denitrifying biofilm"

_PeerJ, doi:10.7717/peerj.4679_

## Round 0.1 · original submission · Minor Revisions

In general I agree with the reviewers, this is an interesting study expanding our understanding of methylotrophic denitrification. While one of the reviewers calls for a major revision, the comments are in general positive. Thus my decision is that the revisions required are rather minor. However, I expect you to address every comment/request carefully.

·

Basic reporting

Basic reporting is ok. Yet, I have some minor comments on figures and tables that should be considered:

Figures:
Please report in the figure caption, if you used SD or SE.
Figure caption for Fig 4 might be too long. Perhaps you could include relevant info from it in materials and methods.
Table 1: annealing T for narG1-primers missing.

Experimental design

no comment

Validity of the findings

Otherwise ok. One thing needs further clarification:
It remains very unclear, why the JAM1 is not reducing nitrite?? Based on Supplementary data 4 and 7 it is expressing Copper-containing nitrite reductase (EC 1.7.2.1). On the other hand in Supplementary document 6, you report that it is not expressing it. Why such discrepancy?
So, do JAM1 actually have gene for nirK-nitrite reductase and it is also expressing it, but the nirK enzyme is not produced or is non-functional?
If JAM1 has nirK gene, what is the similarity of it with that of GP59? Are you then also sure that you really have designed GP59-specific nirK-primers?

Additional comments

General comments:
This is a very interesting and well-done study on Methylophaga strain that can do complete denitrification.
Besides comments above for Basic reporting and Validity of the findings, I have some additional comments that are to be considered.
Abstract:
You could write it more clearly also in the abstract that JAM1 cannot do complete denitrification.
Could you mention something about the transcriptome results in the abstract?

Introduction:
line 58: Besides NO3-, also NO2-, NO and N2O are electron acceptors.
line 78: reference 13 could be positioned later in the sentence.

Materials and methods:
line 114: Why is FeSO4 added two times.
line 117: manufacturer for N2 gas?
line 119: filter-sterilized
line 126: 100 rpm??. could you spesify that they were on a shaker.
line 127-129. Did you work anaerobically during the transfers. How did you measure NO3-?
line 135: saline solution? Can you provide a more specific recipe or name for that??
line 141: How long were the colonies cultured?
line 143: How was NO3/NO2 measured?
line 145: do you mean full length 16S rRNA genes? Please indicate the primer regions or at least the primer names.
line 151: did you secure anaerobic conditions when injecting the cultures to bottles.
line 155. as prescribed? Where?
line 179: did you do size check of the amplicons? E.g. agarose gel electrophoresis.
line 193: Some numbers on the amount of genomic raw data and raw data fragment length could be interesting to see.
Results and discussion:
line 248-250. Why do you think there was this discrepancy with Laurin et al. (23)?
line 303-305. Why does JAM1 have higher dynamism of NO3- processing? Some genetic reasons that can be seen in the genomic comparisons?
line 330. …than IN the overall genome…
line 431. Why do you think there is the strain-specific difference in those encoding the synthesis of osmoprotectant ectoine?
line 440. “etc…” could be removed and the list could be closed without it
line 472: do you mean such accumulation of NO2- and not NO3-?

·

Basic reporting

This manuscript is generally well written, with only a few minor errors in syntax and grammar:
line 29 -- unsure if a culture can be "performed." Maybe inoculated, or set-up, or created
line 62 -- denitrification pathways
line 67 -- bacterial species and numerous protozoans
line 76 -- can carry out complete
line 91 -- lacks a gene
line 101 -- that is closely related
line 119 -- of filter sterilized methanol
line 130 referred to in the text
line 151 -- was added to the medium
line 175 -- was performed
line 227-228 -- rephrase sentence starting, "The intergenic sequences..."

The references and overall structure is clear and well presented. The article has a solid logical flow.

Experimental design

The experimental design is fine -- no comments for revision

Validity of the findings

The data are robust, however, there are a few areas where the significance and rational of the study can be clarified:

line 345 -- perhaps the related Methylophaga species derived from this referenced study can be highlighted in the Figure.

Section starting line 393 -- It is unclear how analysis of CRISPR regions determine whether one strain evolved from another. Can the authors put in a couple of sentences to explain the rationale and the associated references?

Why is the transcriptome data for the denitrification pathways in supplemental data? Also, the text and Table 4 indicate that gene expression in both strains was largely the same, even though the physiology of the strains was quite different. How is this difference reconciled?

The conclusions, particularly from line 474-481, do not reference the data that were collected such as the differences in growth of the isolates or differences (in any) that were found in the transcriptomes. The conclusion should have some reference to how these growth/gene expression parameters could influence enrichment under the conditions in the study. Also, this paragraph does not state why the findings of the particular study are important to further our understanding of methylotrophic denitrifiers. Rather, the conclusions are quite broad and ambiguous and not specific to this work. The authors can improve this last paragraph by stating the major findings of the work and the implications on the study system and the physiology of Methylophaga.

Additional comments

This is a good manuscript with a robust data set that substantially adds to our understanding of methylotrophic denitrifiers.

---

## Round 0.2 · accepted · Accept

I find the revised manuscript much improved. Thank you for your interesting work.

#